# Goal or Miss? A Bernoulli Distribution for In-Game Outcome Prediction in Soccer

**DOI:** 10.3390/e24070971

**Published:** 2022-07-13

**Authors:** Wendi Yao, Yifan Wang, Mengyao Zhu, Yixin Cao, Dan Zeng

**Affiliations:** 1Key Laboratory of Specialty Fiber Optics and Optical Access Networks, Shanghai Institute of Advanced Communication and Data Science, Shanghai University, Shanghai 200444, China; yaowendi1997@shu.edu.cn (W.Y.); yifan@cise.ufl.edu (Y.W.); dzeng@shu.edu.cn (D.Z.); 2School of Computer Science, Fudan University, Shanghai 200433, China; yixin_cao@fudan.edu.cn

**Keywords:** soccer, probability prediction, in-game outcome prediction, machine learning, Bernoulli distribution, regression coefficients

## Abstract

Due to a colossal soccer market, soccer analysis has attracted considerable attention from industry and academia. In-game outcome prediction has great potential in various applications such as game broadcasting, tactical decision making, and betting. In some sports, the method of directly predicting in-game outcomes based on the ongoing game state is already being used as a statistical tool. However, soccer is a sport with low-scoring games and frequent draws, which makes in-game prediction challenging. Most existing studies focus on pre-game prediction instead. This paper, however, proposes a two-stage method for soccer in-game outcome prediction, namely in-game outcome prediction (IGSOP). When the full length of a soccer game is divided into sufficiently small time frames, the goal scored by each team in each time frame can be modeled as a random variable following the Bernoulli distribution. In the first stage, IGSOP adopts state-based machine learning to predict the probability of a scoring goal in each future time frame. In the second stage, IGSOP simulates the remainder of the game to estimate the outcome of a game. This two-stage approach effectively captures the dynamic situation after a goal and the uncertainty in the late phase of a game. Chinese Super League data have been used for algorithm training and evaluation, and the results demonstrate that IGSOP outperforms existing methods, especially in predicting draws and prediction during final moments of games. IGSOP provides a novel perspective to solve the problem of in-game outcome prediction in soccer, which has a potential ripple effect on related research.

## 1. Introduction

Into the 21st century, the sports market has made tremendous progress with the surge of the Internet, especially live streaming and social media. A report by Plunkett Research [1] indicates that the size of the global sports industry is estimated at $1.4 trillion in 2019, within which sports analysis has attracted great attention and investment, bringing many hot topics to researchers. For example, Liu et al. [2] proposed a method to improve basketball dribbling with motion capture data; researchers [3,4,5] applied machine learning in predicting soccer game attendance to serve sales and marketing needs; Lysens [6] was among the first to explore the possibility of predicting health risks in players, and injury prediction has developed rapidly in American professional football [7], baseball [8], basketball [9], and soccer [10,11] leagues.

Soccer is considered the most popular sport in the world and draws attention from both fans and researchers. Valuing Actions by Estimating Probabilities (VAEP) [12] is a groundbreaking framework for valuing any type of player action based on its impact on the game outcome. Bialkowski, A. et al. [13,14] developed a formation detection algorithm equivalent to a constrained K-Means. They discovered roles in a game from data by utilizing a minimum entropy data partitioning method and demonstrated methods to accurately detect and visualize formations. Based on this method, the researchers further designed and developed ForVizor [15], which served as a visual analysis system to support users to conduct formation analysis of the two games.

Predicting game outcomes has been focused on most in sports analysis. Thabtah et al. [16] employed Naïve Bayes, Neural Networks, and Decision Tree Classifiers to predict NBA results. Various feature sets associated with basketball games have been adopted to provide key features for better predictive model performance. Moreover, Chen et al. [17] proposed a hybrid data-mining-based scheme for predicting the final score of NBA games. A two-stage XGBoost [18] model with four pieces of game-lags information achieves the best prediction performance among all competing models. Landers et al. [19] focused on predicting the fantasy points of players and employing these predictions to construct optimal teams. Their model used features such as bookmaker favorites, average (home and away), match location, and other statistics related to team performance. In soccer, Baboota et al. [20] used feature engineering to find feature sets with the most critical factors in predicting game results. Furthermore, the authors used Gaussian Naive Bayes, SVM (with radial basis function and linear kernels), Random Forest, and Gradient Boosting. It was found that the top-performing model was Gradient Boosting, while the worst was Gaussian Naive Bayes.

In soccer, most published studies focus on pre-game outcome prediction, where models utilize features that are available before the game to predict the outcome. Meanwhile, in-game outcome prediction is more challenging and less studied, which refers to the continuous prediction of the outcome based on the changing game state (i.e., score, remaining time, etc.) [21]. In-game outcome prediction is widely conducted in other sports, such as basketball [22,23] and American football [24,25]. However, low-scoring games and frequent draws in soccer make existing methods used in other sports underperform [21]. Low-scoring games make large score differences less common and a lucky goal can put the statistically disadvantaged team in the lead, which makes it more difficult to predict the outcome towards the end of a game. Moreover, soccer games face more frequent draws, which adds more uncertainty.

In addition to the aforementioned problems, in-game outcome prediction requires a large spectrum of real-time information to describe game states, many of which are publicly unavailable. As a result, only a few works have been published on in-game soccer prediction. Zou et al. [26] are among the first to use in-game information in predicting the outcome of soccer games. They estimated a large number of likelihood parameters based on Bayesian methods to model the outcome of a soccer game but failed to elaborate on features used. Klemp et al. [27] found that team strength and number of goals are important contextual factors influencing results within a game analysis framework. They used betting odds as pre-game information and found that pre-game information is more valuable in forecasting soccer games than in-game information in terms of goals. However, instead of making in-game predictions, this research merely used information from the first half to predict the results of the second half. Robberechts et al. [21] also adopted a Bayesian approach to estimate outcomes during the game. They used the Poisson distribution and Auto-Differentiation Variational Inference (ADVI) algorithm [28] to model the number of future goals scored by home and away teams. They also used ten features for each team such as number of goals, remaining time, cards, and expected threat [29] to describe game state. This approach makes consistently correct predictions at the end of each game and has well-calibrated probabilities of win, draw, and loss. However, the parameters of the Poisson distribution are time dependent and difficult to estimate. Furthermore, in this work, the only data used to reflect pre-game competitiveness of both teams are those from ELO ratings [30] from which only limited features can be extracted. The pre-game competitiveness of both teams are thus insufficiently well modelled which then affects the quality of outcome prediction.

This article proposes a two-stage method called in-game outcome prediction (IGSOP) for in-game soccer outcome prediction. It assumes that when the full length of each soccer game is constantly divided into sufficiently small time frames, the number of goals per frame for each team can be represented by a Bernoulli distribution, based on which IGSOP predicts the probability of scoring in each future time frame at the first stage. During the second stage, these probabilities are used to simulate and calculate the number of goals made by each team during the remaining game time. Such calculations, when extensively repeated and the results are statistically analyzed, finally provide predictions for the game. This two-stage method allows for more active handling of impacts from scoring changes. The effectiveness of IGSOP is demonstrated when coupled with the Chinese Super League data, where it outperforms existing methods, particularly in predicting draws and during the final moments of games. IGSOP provides a new perspective for in-game prediction of soccer games.

## 2. Methods

### 2.1. Modeling

As stated in the introduction, most in-game outcome prediction models calculate the probability of outcome directly. This result-based method is prevalent in predictions for basketball and American football, where draws rarely occur. Nevertheless, the frequent occurrence of draws in soccer would seriously reduce the effectiveness of this method [21]. However, an alternative way to predict outcome is to calculate the number of goals to be scored by each team in the future through modeling, before predicting the overall outcome based on these scores. The distribution of the number of goals in a complete soccer game can be simply represented by a bivariate Poisson distribution model [30,31]. As previously demonstrated, Robberechts et al. [21] achieved good results with this method. Nevertheless, in the in-game prediction task, the number of future goals will not be adequately represented by a simple Poisson distribution as game time elapses. All these considered, this paper proposes in-game outcome prediction (IGSOP) to evaluate the in-game outcome by predicting the probability of scoring goals in future time frames.

IGSOP uses a method similar to that in Robberechts et al. [21] but evenly divides the full length of a game into 200 time frames, i.e., 100 time frames for each half. It counts the number of home goals in each time frame under different number splits using data from the Chinese Super League (CSL) between 2012 and 2019 seasons. Figure 1 shows the distributions of number of goals per unit time frame for four different number of time splits. When game time is not divided, the number of home goals in each time frame does fit a Poisson distribution. However, the shorter the time frame is, the fewer the number of possible goals in a time frame becomes. When game time is divided by 40, there are still two consecutive goals in a time frame; at 200, there is at most one goal. The situation then becomes binary: one goal or no goals. A similar observation is seen with the away teams.

As a game is divided into 200 time frames, for each time frame, both the home goal probability $P_{home}\left(x\right)$ and the away goal probability $P_{away}\left(x\right)$ follow the Bernoulli distribution.

First, in a soccer game with a total time frame of T=200, IGSOP estimates the probability sets of scoring for the home team $P_{home,t}$ and the away team $P_{away,t}$ in all future time frames based on the game state $x_{t}$ at time t. The two probability sets at time t can be expressed by the following formula:(1)$$P_{home,t}=\{P_{home}(t+1|x_{t}),\ldots ,P_{home}(200|x_{t}),$$
(2)$$P_{away,t}=\{P_{away}(t+1|x_{t}),\ldots ,P_{away}(200|x_{t}),$$
where $P_{home}(t+1|x_{t})$ and $P_{away}(t+1|x_{t})$ represent the scoring probabilities of the home and away teams in time t+1 based on the game state $x_{t}$.

It is challenging to find all future probabilities of scoring, given that the probability of scoring at each moment cannot be obtained in a real situation. Meanwhile, the ratio of total number of future goals to the length of remaining game can be considered as the mathematical expectation of the goals scored in each time frame. This mathematical expectation is assumed as the probability of scoring in all future time frames.

Although some studies [32,33] have shown that there is a correlation between the goals scored by the two teams, this correlation is weak and thus ignored in this article.

Then, at time t, the number of elements in the probability set for each team is reduced from (T−t) probabilities to one probability. Equations (1) and (2) can be transformed into the following formulae:(3)$$P_{home,t}=\underset{T-t}{\underbrace{\left\{P_{home}(x_{t}),\ldots ,P_{home}(x_{t})\right\}}},$$
(4)$$P_{away,t}=\underset{T-t}{\underbrace{\left\{P_{away}(x_{t}),\ldots ,P_{away}(x_{t})\right\}}},$$
where $P_{home}\left(x_{t}\right)$ and $P_{away}\left(x_{t}\right)$ represent the scoring probabilities of the home and away teams based on the game state $x_{t}$. These two scoring probabilities will be estimated by machine learning models.

Secondly, after obtaining the scoring probability of each team in the future time frame, a Python program is written to simulate the remainder of the game for K times, and corresponding outcome distribution of the game is then calculated. If K is large enough, the number of home team wins, losses and draws can reflect the probability distribution of the outcome. For a game state described by (1) the remaining time $t_{r}$, (2) the home team score $S_{h}$, (3) the away team score $S_{a}$, (4) the probability $P_{h}$ of the home team scoring in each future frame, and (5) the probability $P_{a}$ of away team scoring in each future frame, the specific method for one simulation is as follows: IGSOP distributes $t_{r}$ goal attempts to both teams with probabilities as $P_{h}$ and $P_{a}$ for each attempt, so that the number of goals until end of game by both teams are calculated. These numbers are added, respectively, to the current scores of both teams, to calculate the projected scores by both teams at game end. Such project scores are used to make the win/loss/draw prediction for the underlying simulation. In this paper, we choose K=10,000, which is a large enough number and ensures the simulation efficiency at the same time. This means the above simulation is repeated for 10,000 times while win/loss/draws are counted to provide the final prediction, in the form of probability of each possible outcome.

This approach of estimating game outcomes by simulating goal attempts in each time frame can be used to simulate situations where goal probability is more complex. For the game state $x_{t}$, IGSOP assumes the probability of a team scoring a goal within any future time frame as constant. However, considering the effects of the time and opponent scoring, the goal probability during each time frame can change. It is difficult to calculate the game outcome mathematically due to the variability of probability, but our method can still easily obtain the outcome probability distribution in complex situations.

### 2.2. Flow Framework

The flowchart of IGSOP is shown in Figure 2. The model training phase and the outcome prediction phase both require data processing and feature engineering as essential preprocessing steps. Data processing aims to extract the required information from the vast and complex raw CSL data. Feature engineering further converts the extracted information into features needed by machine learning models to describe game states, as described in Section 2.3. Subsequently, a machine learning model is employed to learn the relationship between the features and the probability of future goals, as explained in Section 3.1. During the outcome prediction phase, the features of the current game state are fed into the trained models to obtain the future goal probabilities in each time frame for each team. Next, IGSOP simulates the remainder of the game with goal probabilities, remaining time, and current score to predict the outcome.

### 2.3. Feature Engineering

In this work, we use the Chinese Super League (CSL) data between 2012 and 2019 seasons collected by China Sports Media, the official partner of the CSL. Such a dataset consists of events manually annotated by professionals from full game video records, supplemented by computer-based quality control. Each event is marked with details such as event type, time, location, and the player who initiated the event.

The quality of the features used for modeling is critical for machine learning tasks. To describe a game state, we divide our features into pre-game features and in-game features. At the beginning of the game, the probability of scoring is mainly determined by pre-game features, such as the competitiveness of both teams. However, as the game goes on, the in-game features gradually become more important.

Baboota et al. [20] propose a series of features for predicting the pre-game outcome in soccer and have achieved promising performance. Such features include the number of shots on target, goals, corners, goal difference to the opposing team, and team ratings. In addition to these, they have also devised three indicators that reflect the team’s recent performance: form, streak, and weighted streak. IGSOP uses almost the same pre-game features. Being slightly different, the team ratings (attack, midfield, defense, overall) for each season are not scraped from the FIFA (International Federation of Association Football) database, as the FIFA database did not record CSL ratings until 2019 season. Three kinds of features are considered to represent the team rating. IGSOP counts the number of goals, goals conceded, and ”streak (as defined in [20])” of each team for each season, and maintains a leaderboard of goals, goals conceded, and streak. The three leaderboards are updated after each round. The ranking of a team on each leaderboard reflects its competitiveness in the corresponding capacity. It is worth noticing that the higher a team ranks on the conceded leaderboard, the worse the defensive ability. In the subsequent modeling and calculations, three competitiveness rating parameters, namely attack rating $R_{attack}$, defense rating $R_{defense}$, and streak rating $R_{streak}$ are introduced to better represent competitiveness and for more convenient data processing. For Team Q, these ratings are defined as:(5)$$R_{attack,teamQ}=\frac{\lambda_{attack,teamQ}-1}{L-1},$$
(6)$$R_{defense,teamQ}=1-\frac{\lambda_{defense,teamQ}-1}{L-1},$$
(7)$$R_{streak,teamQ}=\frac{\lambda_{streak,teamQ}-1}{L-1},$$
where $\lambda_{attack,teamA}$, $\lambda_{defense,teamA}$, and $\lambda_{streak,teamA}$ are the rankings of team Q in the scoring, conceding, and streak leaderboards, respectively and L denotes the total number of teams in the Chinese Football Association Super League. On top of these parameters, IGSOP also adopts a list of features introduced by Baboota et al. [20], which represent pre-game conditions for home and away teams separately, as well as the relationship between the two (the “differential features”). These features have been shown in Table 1.

Furthermore, the in-game features used by IGSOP are primarily derived from event streaming data, which include a detailed and ordered sequence of all player actions during the game. On the CSL data, there are in total 49 event types. However, due to the fact that some event type names have been manually changed and some event types are over-classified, all event types are merged into 22 event types to describe in-game state, including passing, corner, blocking, clearance, dribbles, and shots. All 22 event types presented in Table 2 can provide a comprehensive picture of the state during games. Last but not least, the remaining time of the game and the current score of both teams are the basic and most important features of in-game prediction in soccer.

Events that are far away from the current time have little effect on the goal probability, while the impact of cards will last throughout the game. Therefore, the number of red cards and yellow cards, as well as the moving average number of the remaining 20 events over the latest 20 time frames, are employed to reflect the team’s recent performance. After all features are extracted, the features are normalized to reduce scale differences caused by different physical meanings.

## 3. Results

### 3.1. Prediction Models

We use Python for programming and the Scikit-Learn package to implement machine learning models. To improve performance, we utilize GridSearchCV [34,35], a hyper-parameter automatic search module to search the optimum values of parameters for each model. GridSearchCV runs through all the different parameters that are fed into the parameter grid and evaluates the model for each combination of parameters using the Cross-Validation method.

Excluding some problematic data, there are 1820 games in the CSL during the 2012 to 2019 seasons. In this paper, we use 75% of the data in each season as the training set and the remaining 25% as the test set.

In this task, predicting the probability of scoring goals in each future time frame is a regression problem. To be predicted is the ratio of the number of future goals scored by each team to the length of remaining time. We have tried various regression machine learning models, including Ridge Linear Regression [36], Bayesian Ridge Regression [37], Random Forest [38] and Extreme Gradient Boosting (XGBoost) [18].

We use regression task evaluation metrics Mean Absolute Error (MAE) [39], Root Mean Square Error (RMSE) [39], and R-square (R2) [40] to find the best model. It can be seen in Table 3 and Table 4 that Ridge Linear Regression achieves the best results on home and away team models. Note that lower values of RMSE and MAE are better, while higher values of R2 are better. Therefore, we chose Ridge Linear Regression as a prediction model to predict the probability of the goal.

### 3.2. Method Evaluation

For evaluation purposes, this article compares IGSOP with two other methods. In other sports [23,24,25], result-based methods employ machine learning models for classification to directly predict the probability distribution of the outcome. The result-based method treats the predictions as a classification problem. This paper chooses this multiclass classification (MC) method [23,24,25] as a baseline method. MC is a general method but it will be hampered by the problems associated with the nature of low-scoring games and frequent draws in soccer, as to be discussed in the later experiments. In this paper, MC employs Logistic Regression Classifier [25] to predict final outcomes of games. The target variables of MC are the classification of game outcomes.

The other method is the one proposed by Robberechts et al. [21]. This paper names this method Poisson distribution (PD). It is a goal-based method that first predicts the distribution of goals and then the game outcomes. PD utilizes machine learning algorithms to estimate scoring intensities (the expected value of the Poisson distribution) and uses Poisson distributions to represent the probability of scoring a goal. Next, PD estimates the final outcome based on the future goal distribution of both the home and away teams. PD can better cope with these changes in momentum that often happen after scoring a goal by modeling the number of future goals. However, since the parameters of the Poisson distribution are time-varying, it is difficult to estimate. Slightly different from the method in [21], PD does not use the ADVI algorithm [28] to predict the target variables but the same Ridge Regression algorithm as IGSOP. The target variables of PD are the number of future goals scored by the home and away teams after time t.

Except for the difference in the target variables, all methods are trained with the same set of features. To improve performance, a 5-fold cross-validation grid search is performed for parameter C of Logistic Regression Classifier used by MC and parameter alpha of Ridge Regression used by IGSOP and PD.

We measure the accuracy of our forecasting method with the ranked probability score (RPS) [41,42]. The RPS proved to be more appropriate in assessing probabilistic soccer game predictions than other more popular metrics [43]. The RPS is a scoring function suitable for evaluating the predictions expressed as probability distributions. In soccer outcome prediction, RPS reflects that if the observed outcome is a home win, then a prediction of a draw is more accurate than a prediction of an away win. The RPS represents the difference between cumulative predicted and observed distributions, and is calculated as follows:(8)$$RPS=\frac{1}{r-1}\sum_{i=1}^{r-1}{\left(\sum_{j=1}^{i}\left(p_{j}-e_{j}\right)\right)}^{2},$$
where r is the number of potential outcomes, and  r=3 for home win, draw, and away win in our task. Let $P=\left(p_{1},p_{2},p_{3}\right)$ denote the vector of predicted probabilities for a home win ($p_{1}$), draw ($p_{2}$), and away win ($p_{3}$), and let $E=\left(e_{1},e_{2},e_{3}\right)$ denote the vector of the real results. For the away win E=(0,0,1), if the predicted outcome is P=(0.2,0.2,0.6), then the RPS is 0.1. The smaller the RPS, the better the prediction is.

Table 5 gives the average RPS of the three methods for the first half, the second half, the last 25% of games, the last 10% of games, and the whole game. As shown in Table 5, IGSOP outperforms MC and PD in terms of RPS values. In particularly in the second half of the game, the improvement is more pronounced. In the last 10% of the game, the RPS value of the comparison PD was 58% lower than that of the MC, and the RPS value of IGSOP was 6.4% lower than PD.

The MC method is a pure machine learning approach, predicting game outcomes based on features extracted from current and past states of the game. Current scores by both teams, being treated as merely one of the features by MC, are not given specific consideration. PD and IGSOP, on the other hand, predict goals to be made by both teams during the remainder of the game and adds them to the current scores to make the final prediction. In other words, these two methods put more emphasis on current scores than MC does as the game approaches its end, and thus make better predictions over MC during late games.

To compare in-game outcome predictions, a pre-game outcome prediction model is trained by the XGBoost algorithm with all the pre-game features used by IGSOP. The RPS value of this model on the same test set is 0.2083. This result demonstrates that the added in-game features significantly improve the outcome prediction.

To further demonstrate how significant those differences are, we perform the 5 × 2 cv paired *t*-test [44] to statistically analyze the performance of these methods. The 5 × 2 cv paired *t*-test performs the 2-fold cross-validation 5 times, where, in each repetition, the available data are randomly split into two equal-sized sets. Each of the methods, *A* and *B*, is trained on each of the sets and is tested on the other, correspondingly. In each of the 5 iterations, the 5 × 2 cv *t*-test computes two performance differences:(9)$$p^{\left(1\right)}=p_{A}^{\left(1\right)}-p_{B}^{\left(1\right)},$$
(10)$$p^{\left(2\right)}=p_{A}^{\left(2\right)}-p_{B}^{\left(2\right)},$$
where $p_{A}^{\left(1\right)}$ and $p_{B}^{\left(1\right)}$ represent the performance of methods *A* and *B* on dataset 1, respectively. Then, the mean and variance of the performance differences are calculated the following way:(11)$$\overset{-}{p}=(p^{\left(1\right)}+p^{\left(2\right)})/2,$$
(12)$$s^{2}=(p^{\left(1\right)}-\overset{-}{p})+(p^{\left(2\right)}-\overset{-}{p}),$$

The variance is computed for the five iterations and then used to compute the *t*-statistic as follows:(13)$$t=\frac{p_{1}^{1}}{\sqrt{(1/5)\sum_{i=1}^{5}s_{i}^{2}}},$$
where $p_{1}^{\left(1\right)}$ is the $p^{\left(1\right)}$ obtained from the first iteration. Table 6 shows the calculated *t*-statistic and *p*-values. Considering these values and significance level α=0.05, IGSOP is significantly better compared to MC but not compared to PD. Then, we run a 5 × 2 cv paired *t*-test to compare IGSOP and PD during the last 10% games, and we obtain a *t*-statistic of −2.593 and a *p*-value of 0.048 < 0.05, which means IGSOP does give a better performance than PD during the last 10% games. The final moments of soccer games are often more dramatic, better accuracy leads to greater profitability in both betting and game broadcasting.

At the core of the PD method stands the scoring intensity, which is defined as the expected value of the Poisson distribution that models the number of goals scored by each team in the remainder of the game. The scoring intensity, and the features derived from state of the game that are used to predict it through a machine learning approach, are both time dependent. This introduces great complexity to the algorithm, which grows even greater as the game evolves towards the end as more time-dependent features are involved, and eventually impact the performance of PD during the final moments of games. Test results back up such concern, as the advantage of IGSOP over PD method grows remarkably as the game matures, in terms of prediction accuracy.

For the outcome prediction model, in addition to the gap between the outcome value and the actual value, the calibration of the model needs to be provided to evaluate the performance of the model. In other words, the probability associated with the predicted class label should reflect its ground truth correctness likelihood. For example, when a perfect calibrated model predicts that a draw will occur with 0.3 probability, this means that if 100 games with the same state are played simultaneously, there will be approximately 30 draw games. Our experiments employ the calibration curves [45] and expected calibration error (ECE) [46] as evaluation metrics to assess model calibration. The calibration curve is a visual representation of model calibration. If the model is perfectly calibrated, the calibration curve will be a standard diagonal line. Meanwhile, ECE is a scalar summary statistic of calibration that can compare the difference between two models directly and clearly. To estimate the expected accuracy from limited samples, the predictions are divided into M bins and calculated as a weighted average of the accuracy differences for the bins. The ECE can be computed as:(14)$$ECE=\sum_{b=1}^{M}\frac{n_{b}}{N}\left|acc(b)-conf(b)\right|,$$
where M is the number of bins, which is set to 10 in our experiments. N is the number of samples, and $n_{b}$ is the number of samples in the bth box. acc(b) represents the average value of the actual labels of the samples in the bth box, and conf(b) is the average value of the predicted probabilities in the bth box. The smaller the ECE, the better the calibration.

The calibration curves and ECE of IGSOP, MC, and PD are displayed in Figure 3. From the title of these figures, we can see that the ECE value of IGSOP is 0.02097, which is significantly better than MC and PD. As is shown in these calibration curves, IGSOP curve fits the diagonal excellently in all three cases: home wins, away wins, and draws. The calibration curve of PD also fits the diagonal well but is inferior to IGSOP. Nevertheless, MC fails to perform in predicting draws, as its calibration curve of draws is completely off the diagonal. This is because the machine learning algorithm used by MC directly predicts win/loss of a game instead of actual scores, yet it fails to put extra effort into the draw game situation, which results in a trained model where win/losses are overly represented, and draws are inappropriately under. In contrast, IGSOP calibration curve of draws is the best of these three methods.

### 3.3. Feature Importance

After demonstrating the performance of IGSOP in terms of accuracy and calibration, the impact of the selected features on the prediction results is strongly worth exploring as well. Regression coefficients describe the relationship between predictors and responses in a regression model. To analyze the feature importance, we record the regression coefficients of each feature in the ridge regression model. During the feature engineering phase, all features are normalized to ensure that they are in the same numerical range. This guarantees that the same regression coefficient value for different features represents equal importance under unified standards.

Figure 4 and Figure 5 illustrate the 30 most important features of the home and away teams. It can be discovered that the same features have similar effects on both teams in predicting the probability of scoring a goal. Part of the features and some possible explanations are presented below. For both home and away teams, the number of red cards and the attacking rating greatly affect the prediction of the probability of scoring. A red card will cause a team to reduce a player, and the gap between nine and ten players is enormous, seriously affecting the team’s offense and defense. Interestingly, when one team scores, there is a positive effect on the other team’s offense. It is considered that when one team scores, it will force the other team to attack more aggressively to reduce the impact of the opposing team’s goal. The historical strength of teams, cross, dribbling, and shots in the game all positively impact goals. Clearance is a defensive action, where a player kicks the ball away from his own goal, hence the clearance of the opposing team has a negative effect on goals.

## 4. Discussion and Conclusions

In this paper, we first found that when the full length of each soccer game is divided into sufficiently small time frames, the goal scored by each team in each time frame is modeled by a random variable with a Bernoulli distribution. According to this finding, this paper proposed a novel method named IGSOP for in-game outcome prediction in soccer. IGSOP estimates the in-game outcome in soccer by predicting the probability of a scoring goal in the future time frames, which can achieve the best performance on RPS and calibration compared with existing methods. Since the parameters of the Poisson distribution are time-varying, it brings great complexity to the algorithm, which grows even greater as the game evolves to the end when more time-dependent features are involved, and eventually affect PD’s performance during the late game. This makes the superiority of IGSOP become more prominent during the final moments of games.

The simplicity of IGSOP is appealing. However, to further improve the utility of IGSOP, we perceive that further modifications may be desirable. Our future work will be focused on developing better methods to estimate the probability of scoring goals, which is a time series forecasting problem. Deep learning methods have been proposed as an alternative solution for the time series forecasting problem, where Transformer [47] has been widely employed and achieved excellent performance. Therefore, we shall probably try this method with the hope of achieving a better prediction performance. Further, more detailed and sophisticated features are worth exploring in the future, such as position information and psychological information.

## Figures and Tables

**Figure 1 entropy-24-00971-f001:**
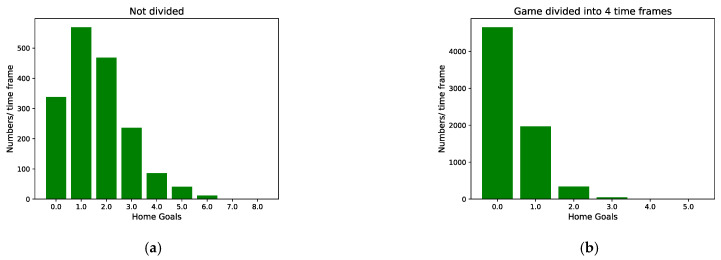
On the 2012–2019 seasons CSL data, the distribution of the number of goals per unit time frame under different split situations: (**a**) no divide, (**b**) one game divided into 4 time frames, (**c**) one game divided into 40 time frames, and (**d**) one game divided into 200 time frames. The numbers on the horizontal axis represent the number of goals scored in a time frame.

**Figure 2 entropy-24-00971-f002:**
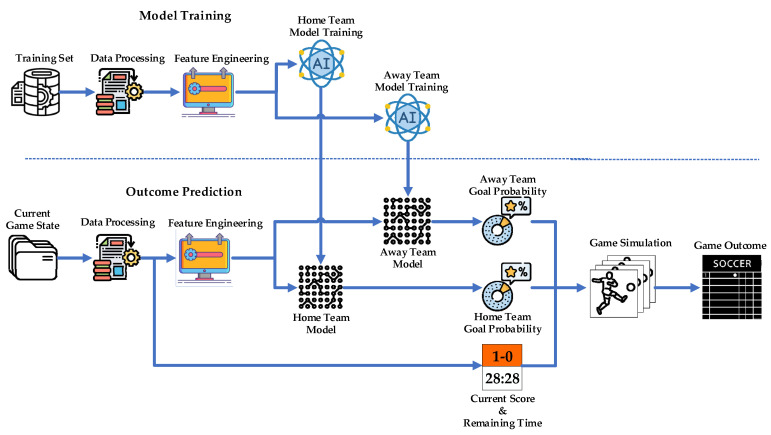
Flowchart of in-game outcome prediction (IGSOP). The upper part represents the training process of the team goal probability model, and the lower part describes the game outcome prediction process.

**Figure 3 entropy-24-00971-f003:**
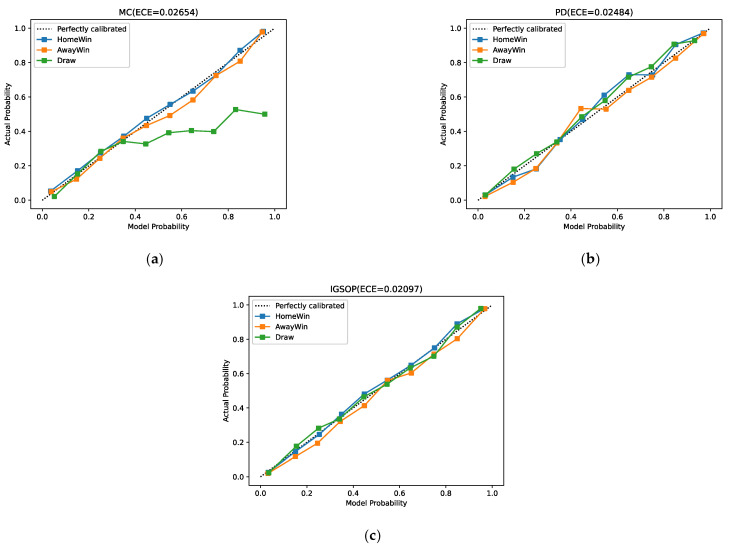
Probability calibration for the multiclass classification method, Poisson distribution method, and IGSOP. (**a**) Multiclass classification calibration curves and ECE, (**b**) Poisson distribution calibration curves and ECE, and (**c**) IGSOP calibration curves and ECE.

**Figure 4 entropy-24-00971-f004:**
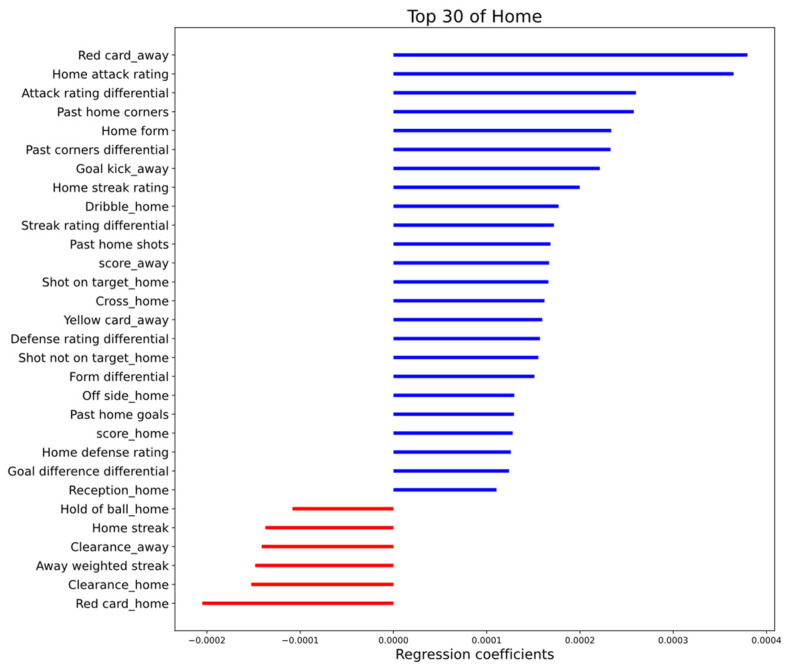
Regression coefficients of the top 30 features that have the greatest impact on the home team’s goal probability. Blue represents a positive effect on the probability of scoring, and red represents a negative effect.

**Figure 5 entropy-24-00971-f005:**
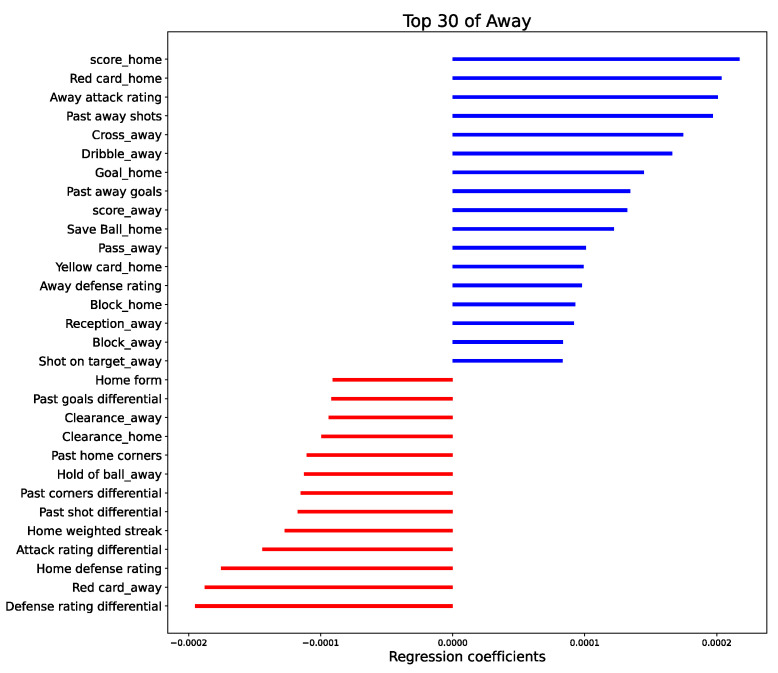
Regression coefficients of the top 30 features that have the greatest impact on the away team’s goal probability. Blue represents a positive effect on the probability of scoring, and red represents a negative effect.

**Table 1 entropy-24-00971-t001:** Pre-game features description.

Home Features	Away Features	Differential Features
Home form	Away form	Form differential
Home streak	Away streak	Streak differential
Past 10 home shots	Past 10 away shots	Past 10 shots differential
Past 10 home goals	Past 10 away goals	Past 10 goals differential
Past 10 home corners	Past 10 away corners	Past 10 corners differential
Home attack rating	Away attack rating	Attack rating differential
Home defense rating	Away defense rating	Defense rating differentia
Home streak rating	Away streak rating	Streak rating differential
Home goal difference	Away goal difference	Goal difference differential
Home weighted streak	Away weighted streak	Weighted streak differential

**Table 2 entropy-24-00971-t002:** Overview of 22 event types.

Event Type	Description
Block	A player blocks a shot on target from an opposing player
Save the ball	A goalkeeper preventing the ball from entering the goal
Chance	A situation where a player should be expected to score
Clearance	A player kicks the ball away from his own goal
Cross	A ball played in from wide areas into the box
Dribble	A player attempts to beat an opponent when he is in possession
Drop of ball	A goalkeeper tries to catch the ball, but drops it from his grasp
Penalty	Foul resulting in a free-kick, penalty, and player out
Hold of ball	A goalkeeper holds the ball in his hands
Own goal	A player kicks a ball into his own net
Pass	Any intentional played ball from one player to another
Reception	Receive the ball from another player
Corner	A kick is taken from the corner of the field
Shot not on target	Shot off the net
Shot on target	Shot into the net, no matter score or not
Tackle	A player takes the ball away from the player in possession.
Free-kick	Direct free-kick and indirect free-kick
Goal kick	The goalkeeper restarts the game and kicks the ball
Goal	Goal and score
Offside	A player who is in an offside position when the pass was made
Yellow card	A player is shown a yellow card
Red card	A player is shown a straight red card

**Table 3 entropy-24-00971-t003:** Comparison of the prediction results of the home team’s goal probability by Ridge Linear Regression, Bayesian Ridge Regression, RF and XGB.

	R2	MAE	RMSE
Ridge Linear Regression	3.695 × 10^−2^	1.619 × 10^−4^	1.272 × 10^−2^
Bayesian Ridge Regression	3.393 × 10^−2^	1.623 × 10^−4^	1.274 × 10^−2^
RF	2.443 × 10^−2^	1.640 × 10^−4^	1.280 × 10^−2^
XGB	3.033 × 10^−2^	1.630 × 10^−4^	1.276 × 10^−2^

**Table 4 entropy-24-00971-t004:** Comparison of the prediction results of the away team’s goal probability by Ridge Linear Regression, Bayesian Ridge Regression, RF and XGB.

	R2	MAE	RMSE
Ridge Linear Regression	1.007 × 10^−2^	1.508 × 10^−4^	1.228 × 10^−2^
Bayesian Ridge Regression	0.995 × 10^−2^	1.508 × 10^−4^	1.228 × 10^−2^
RF	0.135 × 10^−2^	1.521 × 10^−4^	1.233 × 10^−2^
XGB	0.560 × 10^−2^	1.515 × 10^−4^	1.231 × 10^−2^

**Table 5 entropy-24-00971-t005:** Ranked probability score (RPS) of MC, PD, and IGSOP. In addition to the overall RPS, we also calculated the average RPS for the first half, the second half, the last 25% of games, and the last 10% of games.

	First Half	Second Half	Final 25%	Final 10%	Overall
MC	0.1811	0.1099	0.0904	0.0758	0.1455
PD	0.1778 (−1.82%)	0.0913 (−16.9%)	0.0609 (−32.6%)	0.0318 (−58.0%)	0.1346 (−7.49%)
IGSOP	0.1755 (−3.09%)	0.0892 (−18.8%)	0.0570 (−36.9%)	0.0270 (−64.4%)	0.1323 (−9.07%)

**Table 6 entropy-24-00971-t006:** Results of the 5 × 2 cv paired *t*-test performed on MC, PD and IGSOP.

	*t*-Statistic	*p*-Value
IGSOP/MC	−4.512	0.006 < 0.05
PD/MC	−3.270	0.022 < 0.05
IGSOP/PD	−1.689	0.152 > 0.05

## Data Availability

Data sharing not applicable.

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
