# Peer review of "Goal or Miss? A Bernoulli Distribution for In-Game Outcome Prediction in Soccer"

_entropy, 2022, doi:10.3390/e24070971_

Round 1

Reviewer 1 Report

This paper describes a method to produce in-game estimations of soccer results, aiming at guessing the final result (win, draw, loss) at different moments of the game. For this purpose, several historical and in-game features are used to represent the status of each team at every instant of the game, and then machine learning models are used to predict the probability of scoring for the home and away teams, finally leading to an estimation of the result of the game.

This is an interesting topic of research and there is still much work to be done in this field. This paper also analyzes how the different features affect the team’s performance, which leads to very useful insights. Also, dividing the game in short time frames and using a Bernoulli distribution to model the probability of scoring a goal is smart. Nevertheless, I have two comments about this point:

- Equations (1) to (4) need clarification, because it is not clear how you use the probability at different time frames to decide on a final result.

- I don’t see why the authors state that the Poisson distribution is not valid to model in-game predictions. The results presented in this paper prove the opposite, obtaining results that are very similar to those of the proposed method, and also it seems a distribution that can model such events because of its properties. The key is finding the right distribution parameters at each instant of the game.

About the results, the authors state that their method outperforms the other two used for comparison, but they don’t provide information about how significant those differences are. I would recommend the authors to run statistical significance tests and present the results in the paper. Without those, the PD method and the proposed one perform the same. Also, seems like you’re comparing with methods that aim at providing results in sports where the scores are usually higher, therefore the comparison is not fair.

A question about future work: are you planning to use this method to try to predict the exact result of the games? I think that could be another application of such algorithm.

Detailed comments:

- Line 35: add a space between Research and [1].

- Line 68: matching results → I guess you mean match/game results.

- Line 88: didn’t → did not

- Line 88: “the specific features” → not sure what features you’re talking about here.

- Line 88: Klemp et al.[27] → add a space before the reference. Please check this everywhere, there are more.

- Line 103: extra dot at the end of the sentence.

- Line 111: “a host of times” → too colloquial, please change.

- Line 116: “of soccer games in games” → of in-game soccer results?

- Line 124: It’s → It is

- Line 127: achieved results → achieved good results?

- Line 133: the similar method to [21] → a method similar to that in [21]

- Line 137: The paper counts → The paper doesn’t count anything, probably the proposed method does.

- Line 145: A similar goes for away goals → The same is done for away goals.

- Line 152: The numbers on the horizontal axis represent… → Don’t they represent the number of time frames? Please clarify.

- Line 153: “Some bars may not be visible…” → not sure what you mean here, I think you can delete this.

- Line 161: It’s → It is

- Line 177: (1)the remaining → put a space after (1). Same for the rest of the numbers.

- Line 179: extra dot after P_a

- Line 186: the idea behind simulating K penalty shootouts is not pretty clear, please explain in a way that is easier to understand by your audience.

- Line 203: In section 2.4, the machine learning model will be trained → capital S in Section. Please rewrite, since you’re not training the model in Section 2.4

- Line 233: didn’t → did not

- Line 242: not sure what you mean by percentage position. Also, looks like it’s not possible for two teams to have the same R value, and that’s something that should possibly happen.

- Line 257: important features → most important?

- Line 261: rolling average → moving?

- Line 262: remove “type”

- Line 262: over the past 20 time frames → the latest?

- Line 278: IGSOP’s model → remove ‘s

- Line 282: of models → of the models

- Line 289: “to the state that have not appeared before” → to the next states?

- Line 290: this multiclass classification → I guess you mean the method in 23-25.

- Line 294: doesn’t → does not

- Line 302: which is 3 for… → rewrite this sentence, it’s a bit difficult to follow.

- Line 306: same for this sentence: RPS reflects…

- Line 309: table3 → Table 3

- Line 327: model’s performance → performance of the model

- Line 338: can be shown → computed

- Line 340: set as 10 → set to 10

- Line 351: IGSOP’s curve → remove ‘s

- Line 354: do not use “terrible”, too informal.

- Line 355: IGSOP’s calibration → remove ‘s

- Line 377: Whether the home team or the away team → this sounds strange and can be deleted

- Line 397: becomes → become

Author Response

Dear Reviewer:​

Thank you for your letter and for the reviewers' comments concerning our manuscript entitled “Goal or Miss? A Bernoulli Distribution for In-Game Outcome Prediction in Soccer” (entropy-1721614)。 

Those comments are all valuable and very helpful for revising and improving our paper, as well as the important guiding significance to our researches. We have studied comments carefully and have made correction which we hope meet with approval.

The response is in the attachment.

Best Wishes!

Reviewer 2 Report

1. How is the proposed method different from Oracle Cloud Solution that is currently being used in Premier League? As far as I know, Oracle Cloud Solution also provides in-game outcome prediction, which changes in real-time according to various statistical analyses.

https://www.oracle.com/sg/premier-league/ 

2. It is highly recommended to improve the Results section. A detailed description of the competing models must be given along with their advantages and disadvantages. The hyperparameter selection strategy must be explained in detail as well. Currently, the authors simply present the result without providing any in-depth analysis of why certain results were produced.

3. Can you provide the results in a more easy-to-understand way, like accuracy or precision? Since the authors do provide enough information on RPS, it is difficult to understand whether those results in Table 3 are significant.  You can also provide the results of each module, like goal scoring predicting accuracy, and others. After all, you have "Goal or Miss" in the title.

-4. There are several grammatical errors in the paper, along with undefined abbreviations (e.g., ELO). 

- PD doesn’t use ADVI algorithm -> does not

- MC is terrible at predicting draws -> terrible? Overall, I can conclude that the technical writing quality of the paper is poor.

- A player kicks a ball into their own net -> into his own net

- Off side -> Offside

- Stop the ball in flight from scoring -> in flight? What do you mean?

- The ball leaves the player’s controlling area -> Controlling area? I believe that the authors must check all terminologies related to football. 

Author Response

(The authors gave the same response as above.)

Round 2

Reviewer 1 Report

The authors have improved the paper to a great extent, they even added some more experiments that

help understand how the different systems compare to each other, yet there are some aspects that can be improved:

- Line 80: “frequent draws that greatly increases” → frequent draws, which greatly increases

- Line 124: Itis → it is

- Line 153: goals in → goals scored in

- Line 182: atempt → attempt

- Line 201: the machine learning model → a machine learning model

- Line 202: which will be introduced in Section 2.4→ , as explained in Section 2.4

- Line 233: didnot → did not

- Line 258: important → most important

- Line 272: is fed → are fed

- Line 283: Root Means → Root Mean

- Line 284: Table 3 and Table 4, Ridge … → Tables 3 and 4 that Ridge

- Line 296-297 → employ a classification machine learning models to directly predicts… → employ machine learning models for classification to directly predict

- Line 305: predict → predicts

- Line 307-308: and substitutes scoring intesities into Poisson distribution to get the distribution of gaols → and uses Poisson distributions to represent the probability of scoring a goal

- Line 312 → doesn’t not → does not

- Line 319: space before “The RPS”

- Line 320: metrics.[44] → metrics [44].

- Line 322: outcomes predictions → outcome prediction

- Line 353: those differences are. We performe → those differences are, we performed

- Line 356: extra , (sets,.)

- Line 358: difference → differences

- Line 359: extra tabular

- Line 359: where xx and xx represent the performance of methods A and B

- Line 364: extra tabular.

- Line 366: significant different comparing to MC but not significant comparing to PD → significantly better compared to MC but not compared to PD

- Line 373: of the Poisson distribution that goals made during the remainder of game

by each team will follow. → expected value of the Poisson distribution that models the number of goals scored by each team in the remainder of the game

- Line 378: Test result backs such concern → Test results back up such concern

- Lines 458-460: these lines are copy-pasted exactly from another section in the paper. Please don’t do that, rewrite it with different words.

Reviewer 2 Report

I appreciate the authors' replies to my comments.

1. Regarding Response 1: I will take your response, although there are plenty of materials out there that you could use to make a head-to-head comparison.

2. Regarding Response 2: There are so many uncertainties

 - "MC is a general mehod but it will be hampered by the problems associated with the nature of low-scoring and frequent draws in soccer. " -> According to your description, MC predicts the game outcome as win/loss or draw. However, it is not clear to me why MC is hampered by a problem of frequent draws in football? After all, why frequent draws must be even a problem?

- Hyperparameter: You didn't describe what was the hyperparameters (parameters for training the dataset) of the competing methods. 

3. Move Section 2.4 to the Experiment Results section.

4. Regarding Response 4: Once again, I request the authors to solve the proofreading problems. There are still so many typos, grammatically wrong, and awkward sentences. These make paper hard to read and understand. Get some help from a Native Speaker.

Round 3

Reviewer 2 Report

I have no further comments.